The load capacity of maxillary central incisor with simulated flared root canal restored with different fiber-reinforced composite post and cementation protocols

http://orcid.org/0000-0003-2041-9223 Beh Yew Hin 1
Halim Mohamad Syahrizal 2
Ariffin Zaihan 3 zaihan@usm.my
1 Department of Restorative Dentistry, Faculty of Dentistry, Universiti Kebangsaan Malaysia , Kuala Lumpur , Malaysia
2 Conservative Dentistry Unit, School of Dental Sciences, Universiti Sains Malaysia , Kota Bharu, Kelantan , Malaysia
3 Prosthodontics Unit, School of Dental Sciences, Universiti Sains Malaysia , Kota Bharu, Kelantan , Malaysia
Tribst João Paulo
Electronic publication date: 2023 Nov 17
Publication date: 2023
Volume: 11
Electronic Location ID: e16469
Received 2023 Jul 28; Accepted 2023 Oct 25
Copyright: © 2023 Beh et al.
Copyright year: 2023
Copyright holder: Beh et al.
License: This is an open access article distributed under the terms of the Creative Commons Attribution License, which permits unrestricted use, distribution, reproduction and adaptation in any medium and for any purpose provided that it is properly attributed. For attribution, the original author(s), title, publication source (PeerJ) and either DOI or URL of the article must be cited.
License URL: https://creativecommons.org/licenses/by/4.0/

Keywords: Post and core technique, Resin cements, Scanning electron microscopies, Fiber post

Funding: Ministry of Higher Education Malaysia FRGS/1/2021/SKK0/USM/02/31 This work was supported by the Ministry of Higher Education Malaysia for Fundamental Research Grant Scheme with Project Code: FRGS/1/2021/SKK0/USM/02/31 and awarded to Zaihan Ariffin. The funders had no role in study design, data collection and analysis, decision to publish, or preparation of the manuscript.

==============================
Background

This study aimed to evaluate the load capacity of maxillary central incisors with simulated flared root canal restored with different fiber-reinforced composite (FRC) post cemented with either self-adhesive or self-etch resin cement and its mode of fracture.

Methods

Sixty-five extracted maxillary incisors were decoronated, its canal was artificially flared and randomly categorized into group tFRC (tapered FRC post) (n = 22), mFRC (multi-FRC post) (n = 21), and DIS-FRC (direct individually shaped-FRC (DIS-FRC) post) (n = 22), which were further subdivided based on cementation resin. The posts were cemented and a standardized resin core was constructed. After thermocycling, the samples were loaded statically and the maximum load was recorded.

Results

The load capacity of the maxillary central incisor was influenced by the different FRC post system and not the resin cement (p = 0.289), and no significant interaction was found between them. Group mFRC (522.9N) yielded a significantly higher load capacity compared to DIS-FRC (421.1N). Overall, a 55% favorable fracture pattern was observed, and this was not statistically significant.

Conclusion

Within the limitation of the study, it can be concluded that prefabricated FRC posts outperform DIS-FRC posts in terms of the load capacity of a maxillary central incisor with a simulated flared root canal. The cementation methods whether a self-adhesive or self-etch resin cement, was not demonstrated to influence the load capacity of a maxillary central incisor with a flared root canal. There were no significant differences between the favorable and non-favorable fracture when FRC post systems were used to restored a maxillary central incisor with a flared root canal.

Introduction

Special attention and consideration are required when restoring an endodontically treated tooth to its initial form and function; due to its inherent structural weakness post-endodontic treatment (Fráter et al., 2017). It is particularly challenging when dealing with an endodontically treated tooth with a flared root canal. Traditionally, the treatment of choice for a flared root canal is the casted metal-based post and core (Clavijo et al., 2009; Plasmans et al., 1986). This approach requires multiple visits, produces sub-optimal aesthetics, has higher laboratory costs, is technically invasive, and is also associated with a higher degree of catastrophic failures (Gehrcke et al., 2017; Hegde et al., 2012; Zicari et al., 2011).

The introduction of the fiber-reinforced composite (FRC) post system was regarded as a breakthrough. The elastic modulus of the FRC post is close to dentin (13–34 GA), allowing a similar rate of flexion upon loading on the tooth (Asmussen, Peutzfeldt & Heitmann, 1999; Lassila et al., 2005; Sorrentino et al., 2016). It is therefore considered advantageous in reducing catastrophic tooth fracture when compared to the casted metal-based post and core with regards to the long-term survival (Asmussen, Peutzfeldt & Heitmann, 1999; Sorrentino et al., 2016). However, prefabricated FRC posts have a pre-fixed post size, which may require root dentin removal to facilitate its accommodation and fitting into the root canal. The additional removal of root dentin potentially weaken the tooth and reduces its resistance to functional loading (Bittner, Hill & Randi, 2010; Dietschi et al., 2008; Singh, Logani & Shah, 2012). On the other hand, in a flared root canal condition, the opposite occurred, whereby it is difficult to properly fit a prefabricated FRC post. Even the largest available FRC post may not fit the canal well causing poor retention and thick cement space. Hence, various clinical techniques were introduced for the usage of prefabricated FRC post in the flared canal. One of the techniques requires the use of a single FRC post and the remaining spaces were then filled with accessory prefabricated FRC posts (Bonfante et al., 2007; Li et al., 2011). This is to allow more reinforcing fibers to be incorporated into the widely sized canal, subsequently a reduction of cementation space. However, the in-vitro data for this technique in restoring flared root canals was inconclusive (Bonfante et al., 2007; Latempa et al., 2014; Li et al., 2011; Talal et al., 2010).

A novel FRC post system was introduced allowing direct customization known as the direct individually shaped-FRC (DIS-FRC) post, marketed as everStick post by GC Corporation, Japan (Lassila et al., 2005). The unidirectional E-glass fibers are embedded into the uncured resin matrix allowing the application of additional posts which are then molded and shaped according to the final canal configuration. This minimally invasive FRC post system requires minimal to almost no post space preparation, thereby reducing the risk of canal over-preparation (Bell-Rönnlöf et al., 2005).

An optimally bonded FRC post ensures its retention and to creates a uniform distribution of stress along the root when the tooth is in function (Gopal et al., 2017). Self-adhesive resin cement provided a clinically optimal bond strength in bonding the fiber post onto intra-radicular dentin (Bitter et al., 2006; da Silva et al., 2011). This was contributed by the properties of self-adhesive resin cement that demonstrate less sensitivity towards moisture and the ease of clinical handling that consistently produces an optimal adhesion (Goracci et al., 2007). As opposed to the etch and rinse technique, the resultant adhesion can be variable. On the other hand, a self-etch adhesive resin cement exhibits a much more promising bond strength in the intra-radicular environment (Makarewicz et al., 2013). However, much of these data were not established under a flared, large canal configuration where bonding a fiber post with a large cement space can be unpredictable. Other than that, the role of the adhesive system in contributing to the fracture load was not well understood (Sorrentino et al., 2016).

Despite the improvements in the fiber post systems, the debate on the restoration of a flared root canal tooth remains. Therefore, this study aims to investigate: (i) the load capacity of the maxillary central incisor with simulated flared root canal restored with different FRC posts (ii) the load capacity of the maxillary central incisor with flared canals restored cemented with either self-adhesive or self-etch resin cement, and (iii) the mode of fracture of the maxillary central incisors restored with different techniques. The null hypotheses in this study include: (i) there were no difference in the fracture resistance when a maxillary central incisor with simulated flared canal was restored with different FRC post, (ii) there were no differences in fracture resistance when self-adhesive or self-etch resin cement was used for post cementation, and (iii) there was no differences in the resultant mode of fracture.

Materials and Methods

Sample size estimation

The sample size was estimated using GPower version 3.1.9.2 with an effect size of 0.4, a 5% significance level, and a power of 80%. The total required sample size was 66, distributed to three groups.

Samples preparation

This is an in-vitro experimental study that obtained ethical approval from the university’s ethics review board, Jawatankuasa Etika Penyelidikan (Manusia) JEPeM, Universiti Sains Malaysia with protocol code USM/JEPeM/18100577. Written consent was obtained from the eligible patient planning to have their maxillary central incisor extracted due to advanced periodontal disease and other reasons, including tooth extraction due to orthodontic treatment. Extracted sound maxillary central incisors with straight, crack-free roots were collected and stored in 0.9% normal saline at room temperature. The teeth were cleaned with an ultrasonic scaler (Newtron P5XS, Satelec Acteon, Merignac, France) and a number 11 surgical blade (Aesculap, B. Braun, Tuttlingen, Germany). The anatomical crown was sectioned at the cementoenamel junction (CEJ) to produce a root length of 17 ± 1 mm. Its pulp tissue was removed with a barbed broach size ISO 25 (Dentsply, Ballaigues, Switzerland). The roots were then ensured to have a mesio-distal dimension within 5.0–5.5 mm and 7.0–8.0 mm labio-palatally measured at 2.0 mm below the sectioning line for standardization purposes (Hegde et al., 2012). A total of 65 maxillary central incisors were prepared according to the inclusion criteria. The canal flare was simulated using a standardized tapered diamond cutting bur number 103R (Shofu, Kyoto, Japan) which was inserted into the canal up to 7.0 mm depth under copious water irrigation. This resulted in a flared root canal with the widest opening of 2.2 mm in diameter at the coronal opening (Hegde et al., 2012). A shoulder margin of 1.0 mm depth and 2.0 mm height was prepared to resemble a ferrule.

Using a simple random sampling, the samples were randomly categorized into three main groups (Figs. 1A–1C): group tFRC (control), restored with a single tapered prefabricated FRC post (RelyX Fiber Post, 3M Deutschland GmbH, Neuss, Germany); group mFRC restored with multi-FRC post technique consisting of one main parallel-sided prefabricated FRC post (Reforpost, Angelus, Londrina, Brazil) with an additional prefabricated accessory FRC post (Reforpin, Angelus, Londrina, Brazil); and group DIS-FRC restored with a DIS-FRC post (everStick post, GC Europe, Leuven, Belgium). These teeth were further subdivided into two different resin cement: subgroup PC for ParaCore (Coltene, Altstatten, Switzerland) and subgroup RX for RelyX U200 (3M Deutschland GmbH, Neuss, Germany).

Figure 1 Types of fiber posts used.

(A) Single tapered prefabricated FRC post in group 1, (B) parallel prefabricated FRC post with accessory post in multi-FRC post technique in group 2, (C) DIS-FRC post (everStick post) in group 3, and (D) the schematic diagram of experimental setup in universal testing machine.

Post preparation

Group tFRC

Post space was prepared sequentially using RelyX Fiber post drill (3M Deutschland GmbH, Neuss, Germany) up to size 3 drill at 12.0 mm depth. Consequently, a blue coded Ø1.9 mm RelyX Fiber Post (3M Deutschland GmbH, Neuss, Germany) was cut at its coronal end producing a 15.0 mm post length; with 12.0 mm of it in the canal and the remaining 3.0 mm at its coronal end to retain the composite core.

Group mFRC

Post space preparation was done sequentially until the size 5 Largo drill (Angelus, Londrina, Brazil) corresponding to a 1.5 mm post diameter at 12.0 mm depth. One size 3 parallel-sided prefabricated FRC post (Reforpost, Angelus, Londrina, Brazil) with apical diameter Ø1.1 mm and widest diameter Ø1.5 mm was inserted to the desired length followed by a passive addition of one accessory prefabricated FRC post (Reforpin, Angelus Londrina, Brazil) (Bonfante et al., 2007; Talal et al., 2010). The posts were cut to 15.0 mm in length.

Group DIS-FRC

Post space was prepared sequentially using ParaPost drills (Coltene, Altstatten, Switzerland) up to Ø1.25 mm. The canal was then dried with paper points (Meta Biomed, Chungcheongbuk-do, Korea). A Ø1.2 mm everStick post (GC Europe, Leuven, Belgium) was removed from its protective packaging, cut at 15.0 mm length and inserted into the canal. An additional Ø0.9 mm everStick post (GC Europe, Leuven, Belgium) was added to the main post and manipulated in a lateral condensation technique using a finger spreader until the uncured post filled the canal. The bundled DIS-FRC post using everStick post (GC Europe, Leuven, Belgium) was coated with StickRESIN (GC Europe, Leuven, Belgium) and protected from the light source before bonding via the “direct technique” where the post and the resin cement were cured concurrently (Makarewicz et al., 2013).

Cementation protocol

All the canals were irrigated with 2.5% sodium hypochlorite (Pharmaceutical Unit, USM, Kota Bharu, Malaysia) in between instrumentations followed by a final rinse of 17% ethylenediaminetetraacetic acid (EDTA) (Meta Biomed, Chungcheongbuk-do, Korea). All the prepared canals were dried with article points (Meta Biomed, Chungcheongbuk-do, Korea) prior to cementation procedures.

Subgroup PC

Non-rinse conditioner (ParaBond, Coltene, Altstatten, Switzerland) was applied into the canal for 30 s, following which the conditioner was removed and dried with a gentle airstream for 2 s. ParaBond adhesives A and B (Coltene, Altstatten, Switzerland) were mixed in a mixing well and applied into the canal for 30 s using a microbrush. The excess adhesive was removed. Subsequently, ParaCore cement (Coltene, Altstatten, Switzerland) was transferred into the canal with the designated automix tip followed by gentle insertion of the post to its full seating. It was then light-cured for 40 s with a calibrated light-emitting diode (LED) light-curing device (Mini LED, Satelec Acteon, Merignac, France).

Subgroup RX

Self-adhesive resin cement RelyX U200 (3M Deutschland GmbH, Neuss, Germany) was ejected, mixed carefully on a mixing pad, and the post was coated before gently being inserted into the canal, followed by a light-curing of 40 s.

Composite core constructions and thermocyclic ageing

A standardized thermoplastic mold was constructed. A sound maxillary central incisor was prepared with all around shoulder margin with 0.5 mm depth and an axial height of 7.0 mm. A thermoplastic sheet of 1.0 mm thickness (Erkodur, Erkodent, Germany) was vacuum-formed onto the prepared tooth. The standardized composite core was made using the customized mold based on the bonding protocol described for ParaBond (Coltene, Altstatten, Switzerland) using ParaCore (Coltene, Altstatten, Switzerland) resin composite core material. All samples underwent 500 thermocycles at temperatures of 5 °C to 55 °C with a dwell time of 30 s and a transfer time of 5 s using an automated transfer dipping machine (ATDM T6 PD, AMMP Centre, Kuala Lumpur, Malaysia).

Tooth-supporting tissues simulation

The periodontal ligament and alveolar bone simulations were made using the transition wax technique (Soares et al., 2005). Dipping wax (GEO Rewax, Renfert, Hilzingen, Germany) was heated up to 60 °C and the root of each sample was dipped into the wax for 2 s up to 3.0 mm apical to the prepared margin. A hole of 5.0 mm in diameter was created in the center of a transparent plastic film. The specimen tooth was positioned into the hole at 2.0 mm below the CEJ level and was stabilized with wax if needed, ensuring the perpendicular position. The assembly was then embedded into a freshly mixed self-curing acrylic resin (Vertex, Zeist, The Netherlands) with the guide from the transparent plastic film. Upon initial setting of the acrylic, the samples were then removed, and the wax layer was removed entirely. The space created between the root and acrylic was filled with polyvinylsiloxane (Express light body, 3M Deutschland GmbH, Neuss, Germany), simulating the periodontal ligament.

Load-to-fracture test

All the samples were mounted on a designated platform with a specific angle of 135° simulating a class I incisor relationship. Using a universal testing machine (AG-x plus, Shimadzu, Kyoto, Japan), all samples were loaded at a cross-head speed of 0.5 mm/min at 2.0 mm below the incisal edge (Fig. 1D) using a customized cylindrical jig with a diameter of 1.5 mm. The maximum load capacity of the maxillary central incisor was recorded in Newton (N).

Assessment of fracture mode

Under operating binocular loupes with a magnification of 3.0× (SurgiTel, Ann Arbor, Michigan, USA), each sample was assessed and categorized to either favorable fracture (fracture line not extending below the level of the acrylic block, core fracture, post-fracture or post debonding) or unfavorable fracture (fracture line extended below the acrylic level) (Fig. 2) (Hegde et al., 2012). The assessment was performed by two researchers who independently evaluated the samples, and the inconsistent results were resolved by discussion with the third researcher.

Figure 2 Assessment of the mode of fracture.

(A) Favorable fracture, (B) unfavorable fracture, red arrow R, crack/fracture line (note that the sample was lifted from the acrylic block to allow better visualization of the crack extension).

Scanning electron microscopy analysis

Two representative samples from each group that had undergone the load-to-fracture test were randomly selected for further analysis using scanning electron microscopy (SEM). Two grooves were carefully made on the mesial and distal root surfaces. Consequently, the root was split into half, which was then air-dried, sputter-coated with gold powder, and viewed under SEM (Quanta FEG 450; FEI, Lausanne, Switzerland, USA) with secondary electrons 5.00 kV.

Data analysis

The collected data were analyzed using the Statistical Package for the Social Sciences (SPSS) version 26.0 (IBM Corp, Armonk, New York, USA). The descriptive data were analyzed and they met the normality and homogeneity of variances assumptions hence, parametric Two-way analysis of variance (ANOVA) analysis and the least significant difference (LSD) post-hoc test were used to analyze the maximum fracture load of the maxillary central incisor. The mode of fracture was analyzed using the chi-square test for differences. The level of significance was set at α = 0.05.

Results

A total of 65 samples completed the tests and were eligible for data analysis. Due to the limited eligible samples for this investigation, there was a lack of one sample compared to the estimated sample size, leading to an unbalanced samples distribution. Despite that, the final power of this study was 79.78% which was calculated from GPower version 3.1.9.2 software and was considered acceptable. Table 1 summarizes the mean maximum load capacity of the maxillary central incisor and its standard deviation. Following the two-way ANOVA analysis, no interactions were found between the type of post and the type of resin cement in the load capacity of the maxillary central incisor, F (2, 59) = 1.268, p = 0.289 (p > 0.05). However, the load capacity of the maxillary central incisor was significantly affected by the type of posts (p = 0.046, p < 0.05). Therefore, the LSD post-hoc test only applies to the main effect, the post types which revealed a statistical difference between group mFRC (522.9N) and group DIS-FRC (421.1N).

Table 1 Mean maximum load capacity of maxillary central incisor.

Group	n (number of samples)	Load capacity, N (SD)	
tFRC_PC	11	458.2 (91.8)a	
mFRC_PC	10	539.4 (161.3)a	
DIS-FRC_PC	11	437.8 (211.9)a	
tFRC_RX	11	541.5 (119.1)A	
mFRC_RX	11	507.9 (127.5)AB	
DIS-FRC_RX	11	404.3 (79.5)B	
Note:

Different lower case superscript letters representing statistical difference in pairwise comparison using LSD post hoc test when comparing types of post within a single level (subgroup PC). Different upper case superscript letters representing statistical difference in pairwise comparison using LSD post hoc test when comparing types of post within a single level (subgroup RX) whereby p-value < 0.05 is considered significant. Load capacity of the maxillary central incisor in N (Newton). SD = standard deviations.

With regards to the mode of fracture, the percentage of favorable (55%) and unfavorable (45%) fractures were almost equal. Samples in group tFRC_RX demonstrated the most favorable fracture pattern (82%) and this was statistically significant (χ² (1):4.455(1); p = 0.035, p < 0.05) (Table 2). However, other groups did not reveal any significant differences in terms of mode of fractures.

Table 2 Mode of fracture and statistical analysis using chi-square test for differences.

Group	Mode of fracture	χ² (df)	p-value	
Favorable, n (%)	Unfavorable, n (%)	
tFRC_PC	7 (64%)	4 (36%)	0.818(1)	0.366	
tFRC_RX	9 (82%)	2 (18%)	4.455(1)	0.035*	
mFRC_PC	4 (40%)	6 (60%)	0.400(1)	0.527	
mFRC_RX	6 (55%)	5 (45%)	0.091(1)	0.763	
DIS-FRC_PC	4 (36%)	7 (64%)	0.818(1)	0.366	
DIS-FRC_RX	6 (55%)	5 (45%)	0.091(1)	0.763	
Total	36 (55%)	29 (45%)	0.754(1)	0.385	
Note:

*Showing a significant χ² = chi-square test for differences where the significance level was set at p < 0.05.

In the SEM analysis, at the coronal region (Fig. 3) most of the resin cement was observed adhering and covering the post surfaces. An exceptionally thick resin cement was evident in the post in group tFRC, with a sign of cement delamination and cracks exposing the post surface. However, its fibers within the post were seen as attached to its matrix. Other than that, various cracks were observed horizontally and vertically on the cement surface and root dentin. There were gaps and microporosities across most of the samples. Meanwhile, resin tags were observed at the mid-root region (Fig. 4). Its morphology, however, did not exhibit observable differences between the two different types of cement used. The DIS-FRC post revealed signs of fiber detachment from its matrix. At the apical region, resin tags were also present with a higher degree of gaps and microporosities observed (Fig. 5). In this region, more DIS-FRC post fiber detachments were observed, leaving concave surfaces on its matrix known as scalloping.

Figure 3 SEM images at coronal region.

P, post surface; D, dentin surface, red arrow R, crack; blue arrow B, resin tag and green arrow G, gaps and microporosities.

Figure 4 SEM images at mid root region.

P, post surface; D, dentin surface, red arrow R, crack; blue arrow B, resin tag and green arrow G, gaps and microporosities.

Figure 5 SEM images at apical region.

P, post surface; C, cement; D, dentin surface; S, scalloping; H, hackle lines; F, fiber, green arrow G, gaps and microporosities, and blue arrow B, resin tag.

Discussion

Based on the results, different FRC posts resulted in a different maximum load capacity on maxillary central incisors with a flared root canal. This finding was consistent with previous in-vitro studies (Hazzaa, Elguindy & Alagroudy, 2015; Hegde et al., 2012; Kıvanç et al., 2009; Sary, Samah & Walid, 2019; Talal et al., 2010). Therefore, the first null hypothesis was rejected. Group mFRC yielded the highest load capacity value (522.9N), and this technique was previously proven to be comparable to the metal cast post and core in-vitro (Bonfante et al., 2007; Li et al., 2011; Talal et al., 2010). In our study, group mFRC was not significantly different compared to the positive control group tFRC. Meanwhile, group DIS-FRC performed the worst among the groups, with the lowest load capacity recorded (421.1N). These findings were in agreement with the previously reported data (Hazzaa, Elguindy & Alagroudy, 2015; Talal et al., 2010). The contributing factor for this result would be the inferior mechanical properties of DIS-FRC post compared to fully polymerized prefabricated FRC post (Gao et al., 2010). Nevertheless, all the recorded load capacity values were higher than the maximum bite force (80N) at the anterior region in a normal healthy person (Hattori et al., 2009). A significant factor in determining the load capacity is the residual dentin thickness. Thicker root dentin ensures a higher load a particular tooth can sustain (Beltagy, 2017; Bhagat et al., 2017). However, the residual dentin thickness is relatively thin in a flared root canal condition.

Incorporating accessory posts around the main FRC post in group mFRC was to improve its fiber-to-resin composition to enhance the fracture resistance and reduce the resin cement thickness (Latempa et al., 2014; Li et al., 2011; Talal et al., 2010). The increased fiber content optimizes stress distribution within the root canal (Latempa et al., 2014). Although the fiber-to-resin composition in the group DIS-FRC was favorable with minimal cement space (Fráter et al., 2017), an increase in load capacity was not observed in our study compared to the previously reported study (Hazzaa, Elguindy & Alagroudy, 2015; Talal et al., 2010). The low mechanical properties of DIS-FRC post, which exhibits a low elastic modulus and flexural strength, contributed to this phenomenon (Gao et al., 2010). Moreover, the water sorption and hydrolytic degradation between E-glass fibers to the semi-interpenetrating network matrix reduces its flexural strength to up to 35% after thermocycling (Almaroof et al., 2019; Lassila et al., 2005).

The cementation protocols adopted in this study did not influence the load capacity of the maxillary central incisor with a flared root canal. Therefore, the second null hypothesis was accepted. It was previously reported that cementation protocol contributed positively to post retention without exerting a reinforcing effect (da Silva et al., 2015; Dietschi et al., 2008). A simpler cementation protocol, as in the self-adhesive resin cement, ensures minimal flaws during cementation. Hence, a more consistent bond strength can be achieved (da Silva et al., 2015). Interestingly, the main effect analysis in this study indicated that cementation of FRC post with a higher filler content resin cement ParaCore (68% inorganic filler by weight) resulted in no differences in the load capacity among all samples. This observation implies that a high filler content resin cement was favorable in cementing DIS-FRC post as it enhances its mechanical properties. This result is debatable as Alshahrani et al. (2020) did not observe any significant role of filler content in resin cement in terms of increasing load capacity. However, the related samples were not in flared root canals for that study.

Although the overall mode of fracture demonstrates a tendency towards favorable fracture pattern, however our results were not statistically significant. This finding contradicted many other studies which revealed minimal or no root fracture when a tooth was restored with FRC post systems (Beltagy, 2017; Doshi et al., 2019; Fráter et al., 2017; Hegde et al., 2012; Maccari et al., 2007). Nevertheless, our result was in agreement with a few other studies that also observed no significant favorable fracture pattern when a tooth was restored with FRC posts in-vitro (Bell-Rönnlöf et al., 2011; Fokkinga et al., 2006; Fráter et al., 2020; Li et al., 2011; Magne et al., 2017). This was postulated to be due to the thin root dentin in the samples and that the tooth failed prior to the mechanical failure of the post (Maccari et al., 2007). Due to this reason, no incidence of post fracture or debonding was observed in this study. The only exception was in group tFRC_RX, whereby it has a thick cement space with a lowly filled resin cement (self-adhesive resin cement) that acted as a stress breaker and resulted in a higher number of favorable fracture patterns (Hegde et al., 2012; Maccari et al., 2007). It was suggested that the thick layer of resin cement contributed to a relatively weaker bond, causing it to fail preferentially without significant damage to the tooth (Latempa et al., 2014).

Our SEM observation corroborated with Davis et al. (2010) who described the DIS-FRC post fibers detachment from its matrix in an SEM study. The poor coupling and adhesion between the E-glass fibers and the matrix contributed to the reduced physical strength of the post (Davis et al., 2010), leading to an inefficient force transfer and therefore contributing to the lower load capacity on a maxillary central incisor restored with DIS-FRC posts. Due to this disadvantageous feature, other in-vitro studies revealed a higher degree of cohesive failures within the DIS-FRC post (Alnaqbi, Elbishari & Elsubeihi, 2018; Bell-Rönnlöf et al., 2005). However, such an event was not readily observed in the prefabricated FRC post that has better overall mechanical strength compared to DIS-FRC. Despite luting cement delamination and post surface exposure in the samples in group tFRC, the fibers within the post have strongly adhered and well protected by the pre-polymerized highly cross-linked matrix leading to a better outcome on the load capacity of a maxillary central incisor (Davis et al., 2010; Maroulakos, He & Nagy, 2018). The observed gaps and porosities across the dentin-cement and post-cement interfaces were common in SEM studies, probably due to the dehydration protocol for SEM analysis, resin cement polymerization shrinkage or cementation techniques employed during post cementation (Elkassas, Sallam & Ghoneim, 2010; Sarkis-Onofre et al., 2014). The cementation techniques utilized in this study, either using an automix resin cement delivery system delivering resin cement directly into the canal or manually coating the post with resin cement and gently inserted into the canal have resulted in the same observed porosities (Sarkis-Onofre et al., 2014). This SEM study supports the significance of the intimate fiber-matrix interaction as one of the factors contributing to the overall mechanical properties of an FRC post as presented by the prefabricated FRC post. This ensures optimal force distribution and the eventual contribution to the final load capacity of a tooth with flared root canal (Davis et al., 2010).

A large standard deviation was observed in our data, which was a limitation of our study. Natural human teeth are highly susceptible to wear and anatomical variations (Bell-Rönnlöf et al., 2011; Bolay et al., 2012; Yoldas, Akova & Uysal, 2005). Consequently, any inherent micro-cracks on the specimens that were unnoticeable during sample assessments contributed to the large standard deviations (Bolay et al., 2012). On another matter, the collection of maxillary central incisors was challenging due to its aesthetic value. Hence, the eligible samples were less than the required sample size (Bell-Rönnlöf et al., 2011). Another limitation was the loading protocol in this study, which utilized static loading rather than dynamic or cyclical loading. The oral structures were mainly loaded cyclically, and they failed via fatigue stresses (Fráter et al., 2020). The oral cavity, however, is not as simple. In fact, it is so complex and diverse with mechanical forces applied, the thermal changes, chemical changes, pH variations, and the constantly moist environment. Nevertheless, the static load is still valuable as a primary assessment by providing a basic understanding of the biomechanical behavior of dental material (Bell-Rönnlöf et al., 2011). With the basic understanding of the maximum force that a particular biomaterial and the tooth can sustain, a dynamic loading test can be employed to further assess the materials, preventing premature failure when applying a dynamic load (Fokkinga et al., 2006). Endodontic treatment was not performed in this study in view of the possible contamination by the secondary smear layer during gutta-percha removal, which could prevent proper bonding of the post to the root dentin (Bell-Rönnlöf et al., 2011; Elkassas, Sallam & Ghoneim, 2010). Furthermore, endodontic procedures would not affect the load capacity of the samples and, therefore, will not influence the final result (Johnson et al., 2000). Crown restoration was also avoided in this study to prevent confounding effects and to better evaluate the role of a post in the load capacity along with its fracture mode (Annadurai et al., 2019; Beltagy, 2017; Fráter et al., 2017; Kivanç, Alacam & Gorgul, 2010). Since a crown is considered a definitive restoration, the load capacity of a maxillary central incisor may be potentially higher (Annadurai et al., 2019; Beltagy, 2017; Gehrcke et al., 2017; Sary, Samah & Walid, 2019).

Based on the results and limitations of this study, the prefabricated FRC posts, regardless of the techniques employed, suggested promising results compared to DIS-FRC posts in restoring a maxillary central incisor with a flared root canal. However, the role of the DIS-FRC post cannot be entirely omitted. In a situation where customization is desirable, such as in an oval canal, or mildly curved canal or when the coronal fibers require a specific fiber angulation, DIS-FRC can provide a solution (Andhavarapu et al., 2019; Doshi et al., 2019). Moreover, clinical study proves the clinical advantage of DIS-FRC in a large canal with comparable outcomes compared to FRC post and cast gold alloy (Cloet, Debels & Naert, 2017; Zicari et al., 2011). Despite this, more clinical studies with a specific type of teeth are required to define the role of prefabricated or customizable FRC systems in restoring a tooth with flared canal.

Conclusions

Within the limitations of this study, it can be concluded that prefabricated FRC posts outperforms DIS-FRC posts in terms of the load capacity of a maxillary central incisor with a simulated flared root canal. Self-adhesive or self-etch resin cement was not demonstrated to influence the load capacity of a maxillary central incisor with a simulated flared root canal. Hence, there were no significant differences between favorable and non-favorable fracture when various FRC post systems were used to restore a maxillary central incisor with a flared root canal.

Supplemental Information

Supplemental Information 1 Raw data: Load capacity.

Click here for additional data file.

Supplemental Information 2 Raw data: mode of fracture.

Click here for additional data file.

Authors would like to thank those who have directly and indirectly supported the work for this article.

Additional Information and Declarations

Competing Interests

Author Contributions

Human Ethics

Data Availability

The authors declare that they have no competing interests.

Yew Hin Beh conceived and designed the experiments, performed the experiments, analyzed the data, prepared figures and/or tables, authored or reviewed drafts of the article, and approved the final draft.

Mohamad Syahrizal Halim conceived and designed the experiments, analyzed the data, authored or reviewed drafts of the article, and approved the final draft.

Zaihan Ariffin conceived and designed the experiments, performed the experiments, analyzed the data, authored or reviewed drafts of the article, funding acquisition, and approved the final draft.

The following information was supplied relating to ethical approvals (i.e., approving body and any reference numbers):

Universiti Sains Malaysia Human Research Ethics Committee granted an ethical approval with protocol code USM/JEPeM/18100577.

The following information was supplied regarding data availability:

The raw data are available in the Supplemental File.

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
