# Peer review of "The load capacity of maxillary central incisor with simulated flared root canal restored with different fiber-reinforced composite post and cementation protocols"

_PeerJ, doi:10.7717/peerj.16469_

## Round 0.1 · original submission · Major Revisions

Dear Zaihan Ariffin,

I am writing to formally request a major review for your manuscript titled "Load capacity of simulated flared root canal restored with different fiber reinforced composite post" submitted to PeerJ. Please, check the reviewer's comments and enhance the description of the experimental design accordingly. Additionally, improve the sample size and statistical power description, showing how 79.78% was calculated.

**Language Note:** The review process has identified that the English language must be improved. PeerJ can provide language editing services - please contact us at copyediting@peerj.com for pricing (be sure to provide your manuscript number and title). Alternatively, you should make your own arrangements to improve the language quality and provide details in your response letter. – PeerJ Staff

·

Basic reporting

In Background, the authors wrote: This study aimed to evaluate the load capacity of maxillary central incisors……


In Methods: the authors only wrote about load capacity

In Results: the authors only wrote about load capacity and mode of fracture

Note:
There are actually 3 points in this research:
1. the difference in load capacity with different FRC
2. differences in the model of fracture
3. the effect of different cement (self-adhesive or self-etch resin cement) used
In conclusion: the authors should should cover all the three points.

Experimental design

The author states that there are 3 null hypotheses, but the author does not explain what the background/reason for the hypothesis is. The author should write it in the Introduction.
The aim of the study should cover all the points in null hypotheses.

Validity of the findings

How to ensure that the composite cores have the same shape and size for all samples?

The authors states that: …. the types of post which revealed a statistical difference between group 1 (522.9N) and group 3 (421.1N).

In Discussion, the authors should discuss/analyze tthat statement

Additional comments

Please state your conclusion clearly/briefly base on your data/findings.
Note:
There are actually 3 points in this research:
1. the difference in load capacity with different FRC
2. differences in the model of fracture
3. the effect of different cement (self-adhesive or self-etch resin cement) used

Reviewer 2 ·

Basic reporting

There are sufficient updated references.
The article tables and figures are well elaborated.
I think the title must mention the different cementation protocols.
In the first paragraph of the introduction, the authors said that the treatment of choice for a flared root canal is the casted post and core, and they cited two studies of 1986 and 2009, however, later in the second and third paragraphs, different studies of 2005 and 2007 were cited about the use of FRC posts. In literature, many studies indicated the use of FRC posts in the 90s. I think the authors should reconsider that, as many conclusive studies indicated the use of FRC posts much earlier than 2007.
I am not sure I understand what the authors are trying to say in the second paragraph of the introduction, (FRC posts have a pre-fixed post size, which may require excessive root dentin removal to facilitate its accommodation to the root canal…. Unfortunately, even the largest FRC post size may not fit well in flared root canal conditions...). I found these phrases controversial, please try to re-type this paragraph.

Experimental design

The methodology seems to be sound; I just think that the Load-to-fracture test is better than Fracture load testing.
Please reconsider, the largo drill number 5 corresponds to 1.3 mm, and largo drill number 4 corresponds to 1.1 mm, in the text, the authors typed (5 Largo drill….. corresponding to a 1.1 mm)
I think that the authors should cite the articles used to elaborate this methodology.
The manufacturer Reforpost is the same Reforpin, please verify.

Validity of the findings

The results are well presented, however, I found the discussion poor; I think more paragraphs about the study limitations and clinical applicability are highly recommended.

Additional comments

None

Reviewer 3 ·

Basic reporting

1. The manuscript “Load capacity of simulated flared root canal restored with different fiber reinforced composite post” although it present an interesting topic, needs some revision.
2. The English grammar and writing needs revision.
3. The introduction section and the research question could be better explored. The different resin cement/bonding strategies are not even mentioned.

Experimental design

4. What were the reasons for the tooth extractions? That needs to be detailed.
5. How the 65 teeth were distributed among the 6 groups?
6. I suggest changing the group names to a more self-sufficient names, like group 1a changed to FRC_PC, and subsequently. And adding the proper names to the SEM figures. The way they are presented is confusing.
7. Why using a parallel-sided post in group 2? In a flared root canal, it would not make more sense to use a tapered post?
8. What was the reason for the endodontic treatment not be performed in the teeth? Would this interfere the results of the manuscript considering the possibility of pulp remaining in the root canal wall and also considering the dentin modifications due to instrumentation and irrigation?
9. The root canal was dried with paper points only in group 3?
10. How was assured the proper tooth inclination/perpendicular position inside the self-curing resin acrylic? In the same sense, what was the piston used during the fracture load testing? And how the correct positioning of the piston with the tooth was assured?
11. The fracture mode was evaluated by one trained calibrated researcher?
12. The final outcome of the mechanical test was cracks and/or complete fracture of the tooth? Because in the images, only cracks are visible. If only cracks were considered as a final outcome, how the test was limited to that?
13. The SEM specimens were selected after the mechanical test? If yes, this needs to be explained.

Validity of the findings

14. For assuming that the post are or are not protective regarding root fracture the authors should added a control group, without post, and comparing the outcome. Since the authors did not perform such, the paragraph of the failure modes should focus in the tested context.
15. The power analysis should be described in the results section, not in the end of the discussion.
16. One of the main limitations of the study is the absence of a long-term mechanical analysis through fatigue testing. Considering the presence of gaps in the SEM analysis, could the result be different in a fatigue context? This could be explored in the discussion section.

---

## Round 0.2 · Minor Revisions

Dear Authors,

Thank you for submitting your manuscript titled "The Load Capacity of Maxillary Central Incisor with Simulated Flared Root Canal Restored with Different Fiber-Reinforced Composite Post and Cementation Protocols" to PeerJ. After review, we request some minor revisions. First, ensure consistency in stating research objectives between the abstract and the full paper. And provide a more in-depth analysis and discussion of Figures 3, 4, and 5 in the Discussion section. Clarifying these points will strengthen your manuscript for publication.

Kind regards,

·

Basic reporting

1. In the abstract, the research objectives do not match the conclusion. And the objectives in the abstract are not the same as the objectives written in the full paper.

Note:

In Abstract
Background. This study aimed to evaluate the load capacity of maxillary central incisors with simulated flared root canal restored with different fiber-reinforced composite (FRC) post and resin cement.

Conclusion. Within the limitation of the study, it can be concluded that prefabricated FRC posts outperform DIS-FRC posts in terms of the load capacity of a maxillary central incisor with a simulated flared root canal. The cementation methods whether a self-adhesive or self-etch adhesive resin cement, was not demonstrated to influence the load capacity of a maxillary central incisor with a flared root canal. There were no significant differences between favorable and non-favorable fracture patterns when FRC post systems were used to restored a maxillary central incisor with a flared root canal.

Experimental design

In Full Paper

Therefore, this study aims to investigate: i) the load capacity of the maxillary central incisor with flared canals restored with different chairside FRC posts ii) the load capacity of the maxillary central incisor with flared canals restored cemented with either self-adhesive or self-etch resin cement, and iii) the mode of failure of the maxillary central incisors restored with different techniques.


2. In the Discussion, we do not see enough discussion about Figures 3,4 and 5.
And, what can be concluded from Figures 3,4, and 5?

Validity of the findings

-

Additional comments

-

Reviewer 2 ·

Basic reporting

The authors effectively addressed all of my concerns.

Experimental design

The authors effectively addressed all of my concerns.

Validity of the findings

The authors effectively addressed all of my concerns.

Additional comments

None.

Reviewer 3 ·

Basic reporting

The authors have addressed the raised issues. Therefore, the manuscript can be considered for acceptance.

Experimental design

No comment.

Validity of the findings

No comment.

---

## Round 0.3 · accepted · Accept

Dear authors,
I have assessed the revision of the manuscript titled "The Load Capacity of Maxillary Central Incisor with Simulated Flared Root Canal Restored with Different Fiber-Reinforced Composite Post and Cementation Protocols." You have addressed all reviewers' comments effectively. I am confident in stating that the manuscript is now ready for publication.

Best regards,
Dr. Tribst JPM